# Evaluation of the Efficacy and Safety of Quetiapine in the Treatment of Delirium in Adult ICU Patients: A Retrospective Comparative Study

**DOI:** 10.3390/jcm13030802

**Published:** 2024-01-30

**Authors:** Sultan Alghadeer, Rahaf S. Almesned, Emad A. Alshehri, Abdulrahman Alwhaibi

**Affiliations:** 1Department of Clinical Pharmacy, King Saud University, Riyadh 11451, Saudi Arabia; salghadeer@ksu.edu.sa (S.A.); emadab-20011@outlook.sa (E.A.A.); 2Pharmacy Department, King Saud University Medical City (KSUMC), King Saud University, Riyadh 11411, Saudi Arabia; ralmesned2.c@ksu.edu.sa

**Keywords:** quetiapine, haloperidol, olanzapine, risperidone, delirium, QTc interval, ICU, efficacy and safety

## Abstract

**Background:** Quetiapine is commonly prescribed off-label to manage delirium in intensive care unit (ICU) patients. However, limited studies comparing its efficacy and safety to those of other antipsychotics exist in the literature. **Method:** A retrospective, single-center chart review study was conducted on adults admitted to the ICU between January 2017 and August 2022, who were diagnosed with delirium and treated with a single antipsychotic and had no neurological medical conditions, active alcohol withdrawal, or prior use of antipsychotics. Data were analyzed using SPSS software version 28, with *p*-values of <0.05 indicating statistical significance. **Results:** In total, 47 patients were included, of whom 22 (46.8%), 19 (40.4%), 4 (8.5%), and 2 (4.3%) were on quetiapine, haloperidol, risperidone, and olanzapine, respectively. The median number of hours needed to resolve delirium were 12 (21.5), 23 (28), 13 (13.75), and 36 (10) (*p* = 0.115) for quetiapine, haloperidol, risperidone, and olanzapine, respectively, with haloperidol being used for a significantly shorter median number of days than quetiapine (3 (2.5) days vs. 7.5 (11.5) days; *p* = 0.007). Of the medication groups, only quetiapine-treated patients received a significantly higher median maintenance compared to the initiation dose (50 (50) mg vs. 50 (43.75) mg; *p* = 0.039). For the length of stay in the ICU and hospital, delirium-free days, % of ICU time spent in delirium, ventilator-free days, the difference between the highest and baseline QTc intervals, and ICU and hospital mortalities, no significant difference was observed between the groups. **Conclusions:** Overall, the use of quetiapine in our retrospective study seems to not be advantageous over the other drugs in terms of efficacy and safety outcomes.

## 1. Introduction

Delirium is a neurocognitive disorder characterized by a fast and fluctuating disturbance in attention, awareness, and cognition [1]. It has been reported to affect 10–40% of hospitalized patients [2], particularly in critical-care patients, affecting about 80% of mechanically ventilated patients [3]. Delirium can be classified into three subtypes: hypoactive, which is the most frequent subtype among intensive care unit (ICU) patients and usually manifests in a lethargic and calm state; hyperactive, where patients present with agitation and resistance; and mixed delirium, in which patients typically fluctuate between hyperactive and hypoactive states [4,5,6]. Patients who are delirious may experience hallucinations and delusions, as well as confusion and disorientation. In addition to the fear delirium causes in patients, it has been linked to serious consequences such as longer ICU stays and mechanical ventilation, delayed hospital discharge, multiple organ failure, and increased mortality [7,8,9,10].

Pharmacological interventions to treat delirium have been investigated previously, and recent emphasis was placed on this approach after addressing the risk factors of delirium [5,6]. Antipsychotic drugs are commonly used in treating delirium symptoms in clinical practice [3]. In fact, the National Institute for Health and Care Excellence (NICE) guidelines [11] recommend exploring and addressing underlying or reversible causes of delirium, with short-term antipsychotic drugs being considered if distressing symptoms persist. Since then, antipsychotic medications have been widely used to treat delirium in hospitalized patients. A study conducted on data from 300 U.S. hospitals found that around 29% of admissions with delirium received antipsychotics during their hospital stay [12].

Haloperidol is commonly regarded as the “gold standard” antipsychotic for treating delirium due to its minimal anticholinergic adverse effects and low tendency for drowsiness and hypotension [13]. However, it is associated with an increased risk of extrapyramidal adverse effects. Intriguingly, the Pain, Agitation/Sedation, Delirium, Immobility, and Sleep Disruption (PADIS) guidelines [14] for adult ICU patients advise against the routine use of haloperidol for treating delirium. However, despite these recommendations and the associated risks of haloperidol treatment, it continues to be utilized as an anti-delirium agent in the ICU.

Quetiapine, a second-generation antipsychotic, exhibits specific pharmacological properties that make it a potential candidate for the treatment of delirium. It has a high affinity for serotonin, histamine, and a1-adrenergic receptors, as well as a low affinity for dopamine and M1 muscarinic receptors [15]. This unique profile may enable quetiapine to treat delirium and offer sedation while avoiding extrapyramidal adverse effects associated with potent dopamine receptor inhibition, and without inducing delirium via muscarinic receptor inhibition [16]. Several studies have explored the use of quetiapine for the treatment of delirium. One randomized controlled trial (RCT) showed that the addition of scheduled quetiapine to as-needed haloperidol resulted in a shorter time to the first resolution of delirium compared to a placebo with as-needed haloperidol alone [17]. On the other hand, another RCT on medically ill patients demonstrated that low-dose quetiapine and haloperidol were equally effective and safe in managing delirium symptoms [18]. Interestingly, a systematic review of various antipsychotics, including quetiapine, found no significant differences in delirium outcomes. It is important to note that most of the included studies had small sample sizes and varying endpoint measures [19]. However, despite the existing literature, it is still unclear whether the use of quetiapine is advantageous over other antipsychotics, given the inconsistency of the efficacy and safety results published on the treatment of delirium in ICU patients. Our aim in this study was to compare the efficacy and safety of quetiapine versus other antipsychotics for the treatment of delirium in critically ill patients.

## 2. Materials and Methods

### 2.1. Study Design and Setting

This study is a retrospective chart review conducted on patients admitted to the ICU department at King Khalid University Hospital (KKUH)/King Saud University Medical City (KSUMC), a tertiary care teaching hospital located in Riyadh, Saudi Arabia. Data were collected using electronic medical records from January 2017 to August 2022. 

### 2.2. Participants

Adult patients (≥18 years old) who were prescribed any antipsychotic medication during their ICU stay were screened for eligibility for inclusion in our study. Patients who were not delirious, had missing data, were not assessed with the Confusion Assessment Method in the Intensive Care Unit (CAM-ICU) assessment tool (which is valid and reliable to diagnose delirium in ICU patients [20]), had a pre-existing active neurocognitive disease that impacts delirium diagnosis, were diagnosed with alcohol withdrawal syndrome, received multiple antipsychotics for delirium, were scheduled to receive treatment but had not received it, were already taking an antipsychotic prior to admission, or received different strengths of the same medication prior to delirium improvement were excluded from the study (Figure 1). This study was carried out in accordance with the relevant guidelines and regulations (such as the Declaration of Helsinki) [21,22]. It was approved by the institutional review board of KKUH/KSUMC (institutional review board number E-22-6937).

### 2.3. Data Collection

The collected information included: various demographic characteristics, such as age, gender, the primary reason for ICU admission, comorbidities; and clinical characteristics such as the Acute Physiology and Chronic Health Evaluation (APACHE II) score to assess the severity of disease and risk of hospital mortality [23], the Richmond Agitation Sedation Scale (RASS) score to assess the level of consciousness [24], the Glasgow Coma Score (GCS) to assess the level of consciousness impairment [25], the Charlson Comorbidity Index, baseline QTc, use of mechanical ventilation, estimated 10-year survival, and the presence of neurological or neurosurgical conditions. We also examined medication usage during the ICU stay, specifically the utilization of analgesics and sedatives/hypnotics, as well as the potential use of medications that could prolong the QTc interval. Furthermore, information on the duration and dosage of antipsychotic medication use while in the ICU was collected.

### 2.4. Study Outcomes

The primary endpoints included the time to resolve delirium—defined as the time from the initiation of the antipsychotic to the first negative assessments of delirium using the CAM-ICU tool—and duration of antipsychotic use in the ICU. The secondary endpoints included length of stay in the ICU and hospital, % of time spent in delirium while in the ICU, delirium-free days, ventilation-free days, proportion of ICU and hospital mortalities after the initiation of antipsychotics, and the proportion of patients on sedatives/hypnotics while on antipsychotics, along with the duration of their use. The highest QTc interval measurements while on antipsychotics were assessed as a safety endpoint.

### 2.5. Statistical Analysis

Descriptive statistics were used for data analysis. To compare differences between patients treated with different antipsychotics, a non-parametric test was used for continuous variables (median (IQR)), while frequencies and percentages were used for categorical variables. The chi-squared test was used to determine any associations between clinical outcomes and patients’ criteria. The Wilcoxon matched-pairs test was used to assess continuous variable changes within each group separately. Data analysis was performed using SPSS software version 28 (IBM Corp., Armonk, NY, USA). A *p*-value of < 0.05 indicates statistically significant results.

## 3. Results

### 3.1. Demographic Characteristics

Out of 969 electronic medical numbers for patients admitted to the ICU and prescribed any antipsychotic, 125 patients had a confirmed diagnosis of delirium using CAM-ICU. After applying the exclusion criteria, 47 patients were eligible for analysis, of whom 40 were males and 7 were females. Of these, 22 (46.8%), 19 (40.4%), 4 (8.5%), and 2 (4.3%) patients were on quetiapine, haloperidol, risperidone, and olanzapine, respectively (Figure 2). There were several reasons for ICU admission, including cardiac reasons (i.e., coronary artery bypass graft (CABG), aortic valve replacement, cardiac arrest, non-ST-elevation myocardial infarction (NSTEMI), acute decompensated heart failure (ADHF), pulmonary embolism (PE), and sepsis), respiratory reasons (i.e., pneumonia and respiratory failure), trauma (i.e., road traffic accident, falls, and skull fracture), and others (i.e., gunshot, diabetic ketoacidosis, and encephalitis) (Table 1). The baseline characteristics of the patients were generally similar across different medication groups, except for BMI (*p* = 0.027), respiratory diseases as a reason for ICU admission (*p* = 0.022), ischemic heart disease (IHD), and seizure or neurological disorder as comorbidities (*p* = 0.005 and *p* < 0.001, respectively). Further information is provided in Table 1.

Clinical parameters at ICU admission, such as APACHE II score, GCS, Charlson Comorbidity Index, baseline QTc, RASS score, and mechanical ventilation as a marker of severity, were comparable across the medication groups. With respect to the type of delirium, the hyperactive form was seen in 80.8% of the patients. Almost half and 40% of those patients were prescribed quetiapine and haloperidol, respectively. More information about the clinical parameters is provided in Table 2. 

### 3.2. Primary Outcomes

With respect to the median number of days of antipsychotic use, the use of quetiapine was shown to be significantly longer than that of haloperidol (7.5 (11.5) days vs. 3 (2.5) days; *p* = 0.007), while no significant differences were observed between the other groups, as shown in Table 3. When compared to haloperidol, quetiapine was prescribed to a higher number of patients before the onset of delirium, yet the difference was not significant (10 (71.4%) vs. 3 (21.4%); *p* = 0.092). Despite these differences, no significant differences were found in the median time needed to resolve delirium between all groups, including quetiapine versus haloperidol (12 (21.5) hours vs. 23 (28) hours; *p* = 0.115), although the majority of the patients treated with these medications experienced hyperactive delirium. Regarding antipsychotic dose modification, a few patients were started on an antipsychotic after ICU admission but prior to delirium diagnosis and continued on the same antipsychotic therapy after their delirium diagnosis; however, the proportion of those who remained on the same daily prescribed dose even after the delirium diagnosis was 66.6% (2/3) in the haloperidol group, 50% (1/2) in the olanzapine group, and 30% (3/10) in the quetiapine group (Table 3). In spite of that, when the median initiated daily dose (i.e., the dose given either prior to or at the delirium diagnosis) in each group was compared to the scheduled maintenance dose (i.e., the dose given after delirium stabilization), no significant difference was found in the haloperidol group (*p* > 0.99), as shown in Figure 3A; however, the quetiapine-treated patients had received a significantly higher median maintenance dose (50 (50) mg vs. 50 (43.75) mg; *p* = 0.039), as shown in Figure 3B. In other words, among the haloperidol group, six of the nine patients who were included in the comparison were on a maintenance daily dose equal to the initiated dose, which contributed to the absence of significance in the results. Interestingly, when this was investigated in the quetiapine group, 10 of the 19 assessed patients were on the same daily dose, yet the doses of 8 patients were at least doubled, which explains the significant increase in the maintenance dose compared to the initiation quetiapine dose. Due to the insufficient number of patients and missing data, such comparisons could not be conducted for the olanzapine and risperidone groups. Further information is provided in Table 3.

### 3.3. Secondary Outcomes

Similar to the time needed to resolve delirium, all medication groups showed no significant differences in their length of stay in the ICU and hospital, delirium-free days, % of time spent in delirium while in the ICU, ventilator-free days, ICU and hospital mortalities, or use of opiates, sedatives, and hypnotics, nor in their duration of opiate use, as shown in Table 4. Despite the elevated median of highest QTc reading in the risperidone group, no significant change was detected between groups (*p* = 0.304). Additionally, when the difference between the highest and baseline QTc was compared between groups, it was negligible (20.5 (41.25) vs. −21 (0) vs. 36 (53) vs. 39 (35.5); *p* = 0.427, in haloperidol, olanzapine, quetiapine, and risperidone, respectively). More information related to the secondary outcomes is provided in Table 4.

## 4. Discussion

Evidence-based guidelines from organizations such as the American Psychiatric Association, the Canadian Coalition for Seniors’ Mental Health, PADIS, and the United Kingdom’s NICE may preferably emphasize the investigation of non-pharmacological interventions for delirium management [11,14,26,27,28]. In addition, recent randomized clinical trials published after the 2018 PADIS guidelines found that pharmacological interventions, including antipsychotics, may not be considered a usual successful choice for reducing the duration of delirium or improving clinical outcomes [29,30]. However, such trials failed to address the negative sequelae of hyperactive delirium [31]. Evidence-based guidelines recommend short-term usage of antipsychotics for patients with delirium, particularly the hyperactive form [11,14]. Furthermore, about 69% of intensivists reported using antipsychotics for hyperactive delirium [32]. Our results showed that 80.8% of patients presented with hyperactive delirium and, thus, the use of antipsychotics was justified. 

In this retrospective study, the median time needed to resolve delirium was shorter in the quetiapine group compared to the other antipsychotics; however, the difference was not significant (*p* = 0.115). In a systematic review that included 15 studies and 949 patients, nine different antipsychotics were compared with one another and with placebo or usual care. The most frequently used antipsychotics were haloperidol (n = 316), olanzapine (n = 144), quetiapine (n = 125), and risperidone (n = 124) [33]. These commonly prescribed antipsychotics are consistent with the antipsychotics reported in our study. In the systematic study, the authors found that second-generation antipsychotics, in general, had significantly shorter response times compared to the first-generation antipsychotics (SMD = −0.27, 95% CI: 0.54 to −0.01, *p* = 0.04). However, similar to our study, quetiapine as an individual agent did not show such significance. With regards to using quetiapine as an add-on therapy, a randomized controlled trial investigating the effects of quetiapine added to as-needed haloperidol versus as-needed haloperidol alone in 36 patients in an intensive care unit found that adding quetiapine was associated with a shorter time to first resolution of delirium (1 day (IQR, 0.5–3.0) vs. 4.5 days (IQR, 2.0–7.0), *p* = 0.001) [17]. In contrast, quetiapine required a longer time to delirium remission compared to haloperidol (2.6 ± 1.9 versus 1.8 ± 1.5 days) among 52 ICU patients enrolled in a randomized double-blind clinical trial [18]. However, the increase in the time to delirium remission was non-significant (*p* = 0.14). Furthermore, the difference in the time to first remission between the quetiapine and haloperidol groups was also non-significant (HR 1.15, 95% CI: 0.6–2.19, *p* = 0.68). In an observational study conducted in 33 hospitals and with 2834 delirious patients, risperidone (n = 835), quetiapine (n = 779), haloperidol (n = 480), perospirone (n = 88), olanzapine (n = 87), and aripiprazole (n = 61) were the most commonly identified antipsychotics [34]. The rate and duration of delirium among patients using these agents were assessed on a weekly basis. The resolution of delirium within one week or less (≤ one week) was observed in 67% of olanzapine-treated patients, followed by 60% for quetiapine, 56% for perospirone, 52% for haloperidol, 49% for risperidone, and 43% for aripiprazole. Furthermore, olanzapine showed a significantly increase in rate of delirium resolution within one week compared to the others (67% vs. 54%; *p* = 0.025).

In our study, none of the antipsychotics showed any significant differences in terms of length of stay in the ICU and hospital, ventilator-free days, or ICU and hospital mortalities. Our results are consistent with results of a meta-analysis of 19 trials and cohort studies that investigated the efficacy and safety of antipsychotics versus placebo or other antipsychotics among delirious medical, surgical, and ICU patients [35]. The study found no significant association between antipsychotics and the length of hospital stay (n = 1454; mean difference = −0.01 days; 95% CI: −0.16 to 0.14; I2 = 42%), length of ICU stay (n = 1400; mean difference = −0.46 days; 95% CI: −1.15 to 0.24; I2 = 91%), or mortality (n = 1439; OR = 0.90; 95% CI: 0.62 to 1.29; I2 = 0%). In the specific comparison of quetiapine (n = 18) to placebo (n = 18) among delirious ICU patients in a randomized double-blind trial, the length of hospital stay (24 vs. 26 days; *p* = 0.32), length of ICU stay (16 vs. 16 days; *p* = 0.28), and mortality (11 vs. 17%; *p* = 1.0) showed no significant differences [17]. Also similar to our study, there was no difference in the number of ventilator-free days between the two groups (11 vs. 11 days; *p* = 0.67) [17]. The incidence of death was similar between quetiapine (4 deaths out of 21) and the placebo (3 deaths out of 21) in another randomized clinical trial [36]. In another trial comparing quetiapine (n = 24) to haloperidol (n = 28), one patient with malignancy and sepsis died in each study arm [18]. Furthermore, quetiapine failed to improve the length of hospital or ICU stay, or to reduce mortality, when it was used either as a delirium prevention agent in older or high-risk patients, or as a treatment agent for hypoactive delirium [37,38,39].

Initiation and maintenance doses were compared in each group individually in our study. The results showed that quetiapine-treated patients had a significantly higher median maintenance compared to initiation dose while no difference was observed in the haloperidol-treated group. Despite that, it should be noted that the frequency and amount of administered dose of these medications are affected by their natural potency/affinity to dopaminergic receptors, particularly the D2 receptor [40]. In other words, a highly potent antipsychotic such as haloperidol has an overall range of 5–20 mg daily, so ICU patients may require lower doses to control their delirium, which explains the minor differences between initiation and maintenance doses. On the other hand, antipsychotics with lower potency such as quetiapine may require higher doses, particularly higher maintenance doses, which explains the wide overall range (25–800 mg daily) to achieve the same outcomes [30,41]. 

Observable adverse effects such as extrapyramidal symptoms, orthostatic hypotension, or anticholinergic effects seemed to be absent among our study subjects. On the other hand, the QTc interval showed an increase for most of the antipsychotics that were received by our subjects. However, the differences between the baseline QTc intervals and the highest QTc intervals for each antipsychotic were not significant. Despite the fact that second-generation antipsychotics have a lower incidence of extrapyramidal symptoms, neither single nor collective second-generation antipsychotics show significant differences in any individual adverse effect [33]. Additionally, several studies have shown no significant differences in the safety outcomes among various antipsychotics when used for delirium management [33,34,35]. For quetiapine used in delirious patients specifically, it was shown to be associated with a non-significant increase in sedative/sleepiness time compared to haloperidol (33.3% vs. 21.4%; *p* = 0.32) [18]. In addition, there were no differences in any individual adverse effects between quetiapine and the placebo when administered to delirious ICU patients [17]. Overall, there were no significant differences with respect to the use of sedatives/hypnotics or the duration of opiate use between the medication groups.

There are a few limitations to our study. First, the retrospective design of the study inherently poses the drawbacks of missing some data for some patients and normal distribution of the data. Secondly, the overall number of subjects included in the study is a major limitation, which can be attributed to the exclusion criteria that were adopted. However, these criteria were used to ensure a clear investigation of the efficacy and safety of mono-antipsychotic therapy, particularly quetiapine, compared to other antipsychotics. In other words, patients who were not assessed with the CAM-ICU tool as a uniform tool for all participating patients, had a history of neuropsychiatric diseases (including mania, schizophrenia, catatonia, and those who had suicide attempts) that could affect the diagnosis of delirium, were taking an antipsychotic prior to admission, received different strengths of the same medication, or were administered multiple antipsychotics for delirium were excluded. A third limitation is the extremely low number of olanzapine- and risperidone-treated patients, which makes a comparison between the four antipsychotics difficult to interpret. 

## 5. Conclusions

Compared to the other antipsychotics included in our retrospective study, it seems that the use of quetiapine was not advantageous in the time needed to resolve delirium or in the length of ICU/hospital stay. Additionally, the numbers of ventilator-free days and ICU/hospital mortalities were not different between the quetiapine group and the other treatment groups. Quetiapine might not be considered to be the safest and most effective antipsychotic choice for delirium management. However, as with other antipsychotics, its use would be warranted based on its psychopharmacological properties.

## Figures and Tables

**Figure 1 jcm-13-00802-f001:**
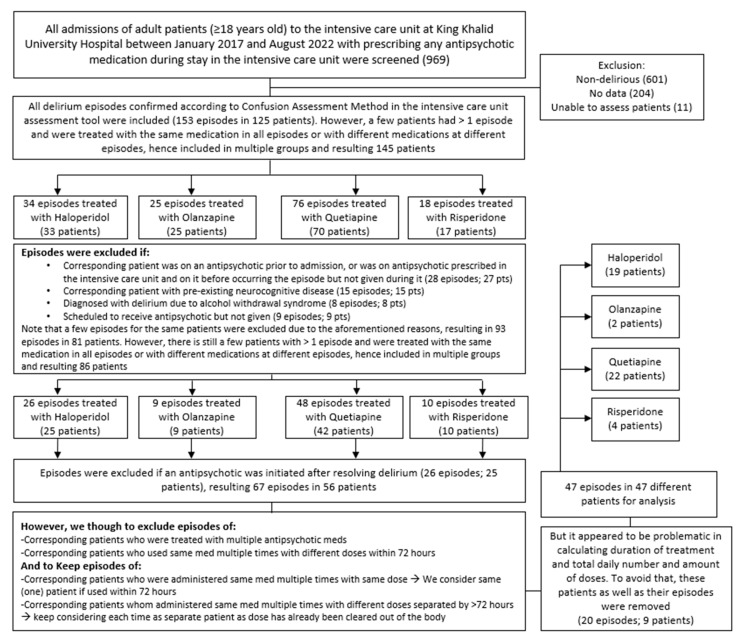
Schematic demonstration of the approach used to include patients in the study (n = 47).

**Figure 2 jcm-13-00802-f002:**
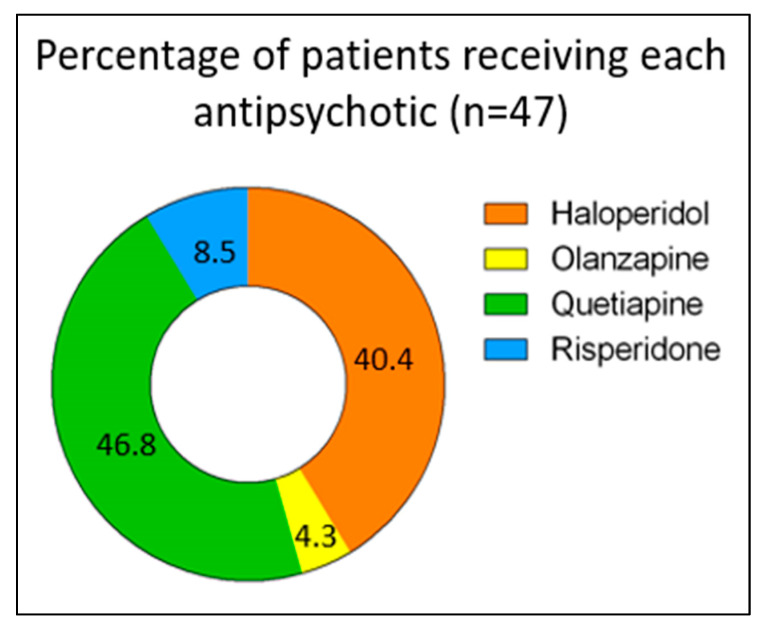
Proportions of patients treated with each antipsychotic (n = 47).

**Figure 3 jcm-13-00802-f003:**
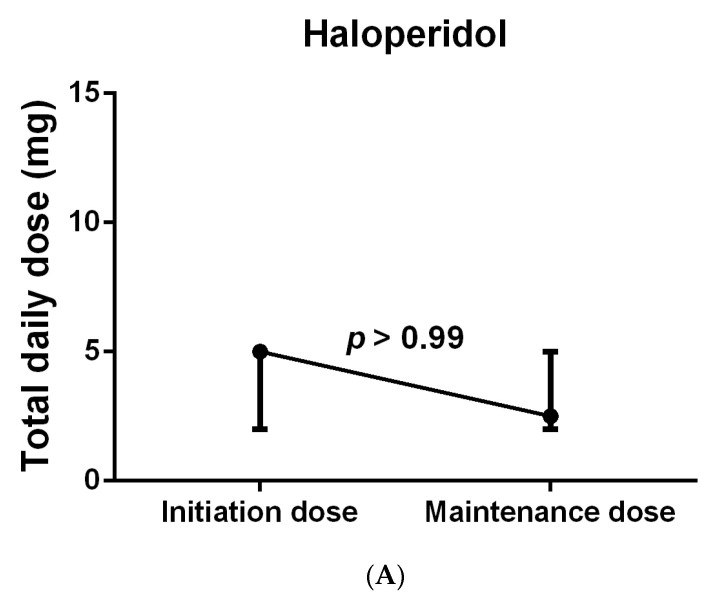
Total daily dose of patients treated with haloperidol (**A**) and quetiapine (**B**). Wilcoxon matched-pairs signed rank test was used to compare initiation and maintenance dose in each group separately. Data were represented as Median (IQR).

**Table 1 jcm-13-00802-t001:** Baseline characteristics (n = 47).

Characteristic	All	Haloperidol (n = 19)	Olanzapine (n = 2)	Quetiapine (n = 22)	Risperidone (n = 4)	*p*-Value
Gender; n (%)						
Male	40 (85.1)	16 (40)	2 (5)	18 (45)	4 (10)	
Female	7 (14.9)	3 (42.9)	0	4 (57.1)	0	0.741
Age (median (IQR)) *	53 (40.5)	59 (31)	54 (4)	35 (37.75)	90 (25)	0.157
BMI (median (IQR)) *	27.5 (8.8)	30.26 (9.4)	31.4 (1.28)	27.3 (8.27)	24 (2.3)	0.027
Reason for ICU admission						
-Cardiovascular	20 (42.6)	11 (55)	1 (5)	5 (25)	3 (15)	0.068
-Respiratory	8 (17)	0	1 (12.5)	7 (87.5)	0	0.022
-Trauma	16 (34)	5 (31.3)	0	10 (62.5)	1 (6.3)	0.398
-Other	7 (14.9)	3 (42.9)	0	3 (42.9)	1 (14.3)	0.870
Comorbidities; n (%)						
-DM	22 (46.8)	11 (50)	1 (4.5)	7 (31.8)	3 (13.6)	0.240
-HTN	20 (42.6)	10 (50)	1 (5)	6 (30)	3 (15)	0.199
-Stroke	2 (4.3)	1 (50)	0	1 (50)	0	0.956
-CHF	9 (19.1)	5 (55.6)	0	2 (22.2)	2 (22.2)	0.172
-IHD	11 (23.4)	7 (63.6)	0	1 (9.1)	3 (27.3)	0.005
-a-fib	4 (8.5)	2 (50)	0	2 (50)	0	0.881
-Asthma	4 (8.5)	3 (75)	0	1 (25)	0	0.513
-CKD	9 (19.1)	5 (55.6)	0	3 (33.3)	1 (11.1)	0.654
-Seizure/neurological disease	1 (2.1)	0	1 (100)	0	0	<0.001
-History of psychiatric disease	2 (4.3)	1 (50)	0	1 (50)	0	0.956
-Consumption of alcohol	3 (6.4)	1 (33.3)	0	2 (66.7)	0	0.869
-Smoking history	10 (21.3)	4 (40)	1 (10)	3 (30)	2 (20)	0.293
-Cancer	4 (8.5)	1 (25)	0	3 (75)	0	0.669
-Other comorbidities	8 (17)	1 (12.5)	0	5 (62.5)	2 (25)	0.119

* Missing data. Abbreviations: BMI (body mass index), DM (diabetes mellitus), HTN (hypertension), CHF (congestive heart failure), IHD (ischemic heart disease), a-fib (atrial fibrillation), and CKD (chronic kidney disease).

**Table 2 jcm-13-00802-t002:** Clinical parameters at ICU admission (n = 47).

Variable	Haloperidol (n = 19)	Olanzapine (n = 2)	Quetiapine (n = 22)	Risperidone (n = 4)	*p*-Value
APACHE II score (median (IQR)) *	17 (11.5)	21.5 (3.5)	15.5 (9.5)	18 (2.75)	0.453
GCS (median (IQR))	11 (11.5)	4 (1)	12.5 (10.25)	14.5 (2.75)	0.387
Charlson Comorbidity Index (median (IQR))	5 (6.5)	1 (1)	0.5 (4)	7 (4)	0.07
Baseline QTc (median (IQR)) *	438.5 (75.8)	442 (0)	445.5 (36.25)	489 (22)	0.143
Mechanical ventilation; n (%)	5 (35.7)	1 (7.1)	6 (42.9)	2 (14.3)	0.718
Estimated 10-year survival % (median % (IQR))	21 (97)	94 (4)	93 (69)	1 (26)	0.588
RASS score during delirium occurrence (median (IQR)) *	1.5 (2)	2 (1)	1.5 (1.25)	1 (1)	0.837
Delirium incidence; n (%)					
-Hypoactive	4 (44.4)	0	4 (44.4)	1 (11.1)	0.892
-Hyperactive	15 (39.5)	2 (5.3)	18 (47.4)	3 (7.9)	
Neurological or neurosurgical condition; n (%)	3 (33.3)	0	6 (66.7)	0	0.438

* Missing data.

**Table 3 jcm-13-00802-t003:** Antipsychotics’ use and impact on delirium (n = 47).

	Antipsychotic				*p*-Value
Variable	Haloperidol (n = 19)	Olanzapine (n = 2)	Quetiapine (n = 22)	Risperidone (n = 4)	
Duration of antipsychotic use in the ICU [days; median (IQR)]	3 (2.5)	12.5 (0.5)	7.5 (11.5)	2.5 (3.25)	0.007 ^#^
Patients on antipsychotics after ICU admission and before the onset of delirium (%)	3 (21.4)	1 (7.1)	10 (71.4)	0	0.092
Time required to resolve delirium while in the ICU and on medication [h; median (IQR)]	23 (28)	36 (10)	12 (21.5)	13 (13.75)	0.115
Use of antipsychotics; n (%)					
-Before and during delirium’s occurrence only	0	0	0	0	0.110
-During delirium’s occurrence only	10 (66.7)	0	3 (20)	2 (13.3)	
-During and after delirium’s occurrence	6 (31.6)	1 (5.3)	10 (52.6)	2 (10.5)	
-Before, during, and after delirium’s occurrence	3 (23.1)	1 (7.7)	9 (69.2)	0	
Dose change of antipsychotic; n (%) *					
-No change	7 (33.3)	2 (9.5)	10 (47.6)	2 (9.5)	0.493
-Increase	1 (11.1)	0	8 (88.9)	0	
-Decrease	1 (50)	0	1 (50)	0	

* Missing data; ^#^ *p* value for comparison of duration of antipsychotic use between haloperidol and quetiapine groups only, while all other comparisons in duration of antipsychotic use between other antipsychotics groups were not significant.

**Table 4 jcm-13-00802-t004:** Overall impact of antipsychotics (n = 47).

Variable	Haloperidol (n = 19)	Olanzapine (n = 2)	Quetiapine (n = 22)	Risperidone (n = 4)	*p*-Value
Length of stay in ICU [days; median (IQR)]	9 (11)	15.5 (7.5)	9.5 (4.75)	4 (5.75)	0.834
Length of stay in hospital [days; median (IQR)]	27 (21)	42 (26)	20.5 (22)	16 (23)	0.388
% of time spent in delirium in ICU [median % (IQR)]	10 (11.85)	21.1 (16.4)	6.25 (12.3)	19.2 (35.4)	0.240
Delirium-free days [days; median (IQR)]	8 (10.13)	13.5 (8.45)	9.3 (6.67)	3.2 (6.8)	0.75
Ventilator-free days [days; median (IQR)]	6 (4)	2 (0)	4 (5.5)	6 (4)	0.58
New ICU mortality; n (%)	1 (25)	0	2 (50)	1 (25)	0.604
New non-ICU mortality; n (%)	1 (33.3)	0	2 (66.7)	0	0.869
Overall mortality; n (%)	2 (28.6)	0	4 (57.1)	1 (14.3)	0.766
Use of opiates; n (%)	11 (44)	2 (8)	9 (36)	3 (12)	0.259
Duration of opiate use; [days; median (IQR)] *	1.5 (1)	1.5 (0.5)	1 (0)	1.5 (0.5)	0.666
Use of BDZ; n (%)	9 (45)	1 (5)	7 (35)	3 (15)	0.394
Use of dexmedetomidine; n (%)	5 (31.3)	1 (6.3)	10 (62.5)	0	0.254
Use of propofol; n (%)	3 (75)	0	1 (25)	0	0.513
Use of clonidine; n (%)	3 (25)	0	9 (75)	0	0.125
Highest QTC while on antipsychotics [median (IQR)] *	466 (52)	421 (0)	477 (65)	527 (76.5)	0.304

* Missing data.

## Data Availability

The data presented in this study are available upon request from the corresponding author. The data are not publicly available due to privacy reasons.

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
