# Peer review of "Evaluation of the Efficacy and Safety of Quetiapine in the Treatment of Delirium in Adult ICU Patients: A Retrospective Comparative Study"

_jcm, 2024, doi:10.3390/jcm13030802_

Round 1

Reviewer 1 Report

Comments and Suggestions for Authors

This study explores the efficacy and safety of an atypical antipsychotic, i.e., quetiapine, compared to other three antipsychotics, in the treatment of delirium in adult ICU patients. The topic is interesting and the study results may have important practical consequences for the case management of these patients. Please refer to the following observations:

Proofreading is necessary because existing grammar and syntactic errors must be corrected, as some of them change the significance of the sentences.

Line 22- all acronyms should be explained, like in this case „IHD”; the same for „LOS” in line 32;

Lines 31-32- did the Authors mean that no significant difference was detected between quetiapine and risperidone? What about the comparison between quetiapine and other antipsychotics used on the QTc interval?

Lines 34-36- it is unclear what the Authors mean by „exciting evidence related to the superiority of quetiapine” since the current study is negative;

Line 37- consider adding more keywords, in order to make the article easier to be discovered by potential readers (for example, risperidone, haloperidol, QTc interval)

Lines 48-50- It is important to address the fact that pharmacological interventions are recommended after risk factors for delirium have been explored and treated- https://doi.org/10.1186/s42466-021-00110-7,  https://pubmed.ncbi.nlm.nih.gov/23269131/, etc.

Fig.1- please specify the entire duration of data collection, not only the start date (2017);

Table 3- please specify the measure unit used for “Use of antipsychotic while ICU admission…”;

Line 277- please remove this sentence fragment, because it looks like a remnant of the manuscript’s template.

Comments on the Quality of English Language

Proofreading is required.

Author Response

Comment 1: Proofreading is necessary because existing grammar and syntactic errors must be corrected, as some of them change the significance of the sentences

Response 1: We appreciate the reviewer for bringing this to our attention. English editing service from MDPI was used to correct grammar mistakes and improve English language of the manuscript.

Comment 2: Line 22- all acronyms should be explained, like in this case „IHD”; the same for „LOS” in line 32

Response 2: We thank the reviewer for this comment. All acronyms were explained as requested

Comment 3: Lines 31-32- did the Authors mean that no significant difference was detected between quetiapine and risperidone? What about the comparison between quetiapine and other antipsychotics used on the QTc interval?

Response 3: We appreciate the reviewer for this question. When mean of highest QTc interval was compared between the groups, no significant difference was found when any group was compared to others except when risperidone group only was compared to haloperidol group. We have removed that to adhere to the word limits in the abstract however explained that in details in the results section.

Comment 4: Lines 34-36- it is unclear what the Authors mean by „exciting evidence related to the superiority of quetiapine” since the current study is negative

Response 4: We appreciate the reviewer for bringing this typo to our attention. We meant “existing evidence”. We have removed that to adhere to the word limits in the abstract.

Comment 5: Line 37- consider adding more keywords, in order to make the article easier to be discovered by potential readers (for example, risperidone, haloperidol, QTc interval)

Response 5: We appreciate the reviewer for this suggestion. New keywords include “Quetiapine; Haloperidol; Olanzapine; Risperidone; Delirium; QTc interval; ICU; Efficacy and Safety”

Comment 6: Lines 48-50- It is important to address the fact that pharmacological interventions are recommended after risk factors for delirium have been explored and treated- https://doi.org/10.1186/s42466-021-00110-7,  https://pubmed.ncbi.nlm.nih.gov/23269131/, etc.

Response 6: We appreciate the reviewer for this comment. We have included the following sentence to highlight this point: “Pharmacological interventions to treat delirium have been investigated previously and recent emphasis was made on this approach after addressing risk factors of delirium (PMID: 23269131; https://doi.org/10.1186/s42466-021-00110-7).”

Comment 7: Fig.1- please specify the entire duration of data collection, not only the start date (2017)

Response 7: We appreciate the reviewer for this suggestion. We have modified that to “between January 2017 and August 2022” and updated the figure accordingly.

Comment 8: Table 3- please specify the measure unit used for “Use of antipsychotic while ICU admission…”;

Response 8: We appreciate the reviewer for this comment. We have changed this to “patients on antipsychotic after ICU admission and before onset of delirium (%)”

Comment 9: Line 277- please remove this sentence fragment, because it looks like a remnant of the manuscript’s template.

Response 9: We really appreciate the reviewer for this note. We have removed it.

Reviewer 2 Report

Comments and Suggestions for Authors

The paper by Sultan Alghadeer et al. is a retrospective study evaluating efficacy and safety of quetiapine in the treatment of delirium compared to other anti-psychotics. The study design (i.e. retrospective) is not the most suitable for evaluating the efficacy of a treatment and the sample size, also in relation to the strict inclusion and exclusion criteria, is rather small, making the results scarcely generalisable. Furthermore, I have several suggestions for the Authors: 

- Please always write the extended word first and then its acronym;

- In the Introduction, please elaborate on the subtypes of delirium, particularly the one on which most analyses then focus;

In the Methods:

 - please provide a brief explanation of the measuring instruments used (e.g. CAM-ICU, GCS, RASS);

- please remove the abbreviations in Figure 1.

- please indicate whether the study is in accordance with the Declaration of Helsinki by citing the relevant bibliography; 

- Line 92: Does this refer to patients who have received anti-psychotic medication for causes other than delirium? E.g. for what causes?

- Please provide more specifically the reasons for admission to the ICU and those, if possible, that led to delirium;

- Statistical analysis: how was the normal or non-normal distribution of the data assessed? How was the marked inhomogeneity in terms of sample size between the different subgroups considered?

- The Authors also examined patients admitted during the period of the Covid-19 pandemic. Were any of them (at the time of admission or not) infected by Sars-Cov2? Or had they received vaccination (one or more doses?)? If so, is there any relation to the otucomes under investigation? Several studies have examined the role of Covid-19 in the pathogenesis of delirium in hospitalised patients; 

- In Table 1, please provide a legend for the acronyms;

- Lines 163-165: Given the exclusion criteria reported by the Authors, the meaning of this sentence is not clear to me;

- Discussion: the results of the study should be further elaborated by considering the sample limits for each group and the covariates of interest, instead summarising the results of the other works reviewed. 

 - Line 277: to be removed. 

Author Response

Comment 1: The paper by Sultan Alghadeer et al. is a retrospective study evaluating efficacy and safety of quetiapine in the treatment of delirium compared to other anti-psychotics. The study design (i.e. retrospective) is not the most suitable for evaluating the efficacy of a treatment and the sample size, also in relation to the strict inclusion and exclusion criteria, is rather small, making the results scarcely generalisable.

Response 1: We thank the reviewer for this valuable comment. We have stated these as limitations for our study

Comment 2: Please always write the extended word first and then its acronym

Response 2: We appreciate the reviewer for this comment. We have considered this suggestion throughout the manuscript.

In the Introduction:

Comment 3: Please elaborate on the subtypes of delirium, particularly the one on which most analyses then focus

Response 3: We appreciate the reviewer for this comment. We have elaborated on the subtypes as requested but we didn’t focus on any type as it was not considered among the aims that we did set for the study.

In the Methods:

Comment 4: Please provide a brief explanation of the measuring instruments used (e.g. CAM-ICU, GCS, RASS)

Response 4: We appreciate the reviewer for this suggestion. We have briefly explained each and provided references.

Comment 5: Please remove the abbreviations in Figure 1.

Response 5: We appreciate the reviewer for this suggestion. All abbreviation were removed and fig was updated accordingly.

Comment 6: Please indicate whether the study is in accordance with the Declaration of Helsinki by citing the relevant bibliography

Response 6: We appreciate the reviewer for this suggestion. We indicated that in the method as requested.

Comment 7: Line 92: Does this refer to patients who have received anti-psychotic medication for causes other than delirium? E.g. for what causes?

Response 7: We appreciate the reviewer for this question. Since we were only able to retrieve information for ICU patients through administered medications, we expected to encounter cases where patients are on antipsychotics for neuropsychiatric illnesses. There were some patients who had established diagnosis of mania, schizophrenia, catatonia and other psychotic conditions and those who had suicide attempts. These patients were not included in our study.

Comment 8: Please provide more specifically the reasons for admission to the ICU and those, if possible, that led to delirium

Response 8: We appreciate the reviewer for this suggestion. There are several reasons for ICU admissions including cardiac reasons (Post CABG, valve replacement, post cardia arrest, NSTEMI, acute decompensated heart failure, PE, sepsis, etc), respiratory reasons (pneumonia, respiratory failure, etc), trauma (road traffic accident, falls, skull fracture, etc) and others (gunshot, diabetic ketoacidosis, encephalitis, etc). These modifications are incorporated in the manuscript.

Statistical analysis:

Comment 9: how was the normal or non-normal distribution of the data assessed? How was the marked inhomogeneity in terms of sample size between the different subgroups considered?

Response 9: We appreciate the reviewer for these questions. Given the limited number of patients and retrospective nature of the study, we expected to have such disturbance in distribution of the data. We have included that as a limitation of the study.

Comment 10: The Authors also examined patients admitted during the period of the Covid-19 pandemic. Were any of them (at the time of admission or not) infected by Sars-Cov2? Or had they received vaccination (one or more doses?)? If so, is there any relation to the otucomes under investigation? Several studies have examined the role of Covid-19 in the pathogenesis of delirium in hospitalised patients

Response 10: We appreciate the reviewer for these questions. There are only 3 of the included patients had confirmed diagnosis of COVID-19 (2 in the quetiapine group and 1 in the olanzapine group), yet no information about vaccination were collected. We are unable to disclose further details as another project on delirium among COVID-19 patients is currently conducted by our team.

Comment 11: In Table 1, please provide a legend for the acronyms

Response 11: We appreciate the reviewer for this suggestion. We have included this as requested.

Comment 12: Lines 163-165: Given the exclusion criteria reported by the Authors, the meaning of this sentence is not clear to me

Response 12: One of the exclusion criteria was “already taking an antipsychotic prior to ICU admission”. However, some patients were started on an antipsychotic before confirming delirium diagnosis using CAM-ICU and continued/scheduled on the same medication after delirium stabilization. In details, 3 in haloperidol group (two of them continued on the same dose), 2 in olanzapine group (1 continued on the same dose) and 10 in quetiapine group (3 continued on the same dose). Given the limited number of patients, we have included these and analyzed their impact on delirium. Despite the thoughts that high number of quetiapine-treated patients were on it before delirium diagnosis might potentially reduce the time of delirium, this was not the case as shown in table 3. In fact, an increase in dose was required in maintenance therapy beside the long duration of treatment time of as shown in figure 2b, and in table 3, respectively. This could put the patients at higher risk of adverse effect. This was explained in thoroughly in lines 185-200 in the updated version.

Discussion:

Comment 13: the results of the study should be further elaborated by considering the sample limits for each group and the covariates of interest, instead summarising the results of the other works reviewed.

Response 13: We appreciate the reviewer for this suggestion. We would like to clarify that in the discussion section, we aimed to cover all related subjects regarding the delirium and the use of antipsychotics, with focusing on our results for the primary and secondary outcomes of the current study. Different readers may have different preferences. The extensive summarization of similar results in other literature or guidelines and link to the current study results could be of interests for most readers. Thus, we started the discussion discussing the various international guidelines that highlight the use/recommendations of pharmacological intervention, particularly antipsychotics, and how such recommendations related to our study results. Then in each paragraph, we summarized the results of our study (primary and secondary outcomes that include "the time to delirium resolution", "length of stays in ICU and hospital", "ventilator free days", "ICU and hospital mortalities", and "adverse effects") with comparing these results with other related published studies. Finally, we ended up the discussion section with the limitations. 

Comment 14: Line 277: to be removed.

Response 14: We really appreciate the reviewer for this suggestion. Line 277 was removed as requested.

Reviewer 3 Report

Comments and Suggestions for Authors

There are some revisions that I would like the authors to address and/or to consider.

1.      Please reformat the abstract according to the journal format

2.      Please justify the reason for the conducted study

3.      study design and sample size are not appropriate and large enough to make conclusion

4.      discussion section has not been well developed

Author Response

Comment 1: Please reformat the abstract according to the journal format

Response 1: We appreciate the reviewer for this comment. Abstract was reformatted according to the journal format.

Comment 2: Please justify the reason for the conducted study

Response 2: We appreciate the reviewer for this comment. The reason behind conducting this study was stated in the last paragraph on the introduction, as the following “despite the existing literature, it is still unclear whether use of quetiapine is advantageous over other antipsychotics given the inconsistency of efficacy and safety results published on the treatment of delirium in ICU patients” (Line 76-81)

Comment 3: Study design and sample size are not appropriate and large enough to make conclusion

Response 3: We appreciate the reviewer for this comment. We have stated that as a limitation of our study.

Comment 4: Discussion section has not been well developed

Response 4: We appreciate the reviewer for this comment. We have reviewed the discussion and edited in order to improve it.

Round 2

Reviewer 1 Report

Comments and Suggestions for Authors

Thank to the Authors for considering my recommendations, but there are a few aspects that require further clarification, please see below:

Lines 22 and 184-185- Please specify the units used for the measurement of the initiation dose;

Where are the figures 2a and 2b? In the manuscript, there is only one figure 2 and two figures 3 (a and b). Please remove the symbol after  A and B in Figure 3.

Line 96- What does „received different strengths of the same medication” mean? Did the Authors refer to variations in the daily doses in the same patient during the ICU stay? Or maybe to the patients who received different daily doses prior to the inclusion in the current study? Please clarify.

Table 1- What type of psychiatric disorders did the patients present?

Table 2- The comparison between the four antipsychotics is difficult to interpret, because only 4.3% and 8.5% of the patients received olanzapine and risperidone, respectively; this is a limitation of the study that should be mentioned in the dedicated paragraph;

Lines 174-176- Why were some patients initiated on antipsychotics after ICU admission but prior to the delirium diagnosis? Did they have psychotic disorders or other pathology requiring antipsychotics? According to the exclusion criteria in lines 90-97, patients who were not delirious were not included in the analysis. Clarifying this aspect is important since 21.4% of the haloperidol-treated patients and 71.4% of the quetiapine-treated patients were in this category.

Lines 185-193- those differences between the initial and maintenance doses of haloperidol and quetiapine, respectively,  may be related to the distinct pharmacology of the two antipsychotics, because haloperidol is a strong antagonist of D2 receptors and quetiapine only a weak D2-antagonist; increasing the dose of quetiapine is perfectly explainable because at low doses it is only used as an off-label hypnotic (25-200 mg/day) -https://pubmed.ncbi.nlm.nih.gov/22510671/; also, the admitted interval of doses for haloperidol according to SPC is 5-20 mg/day, and for quetiapine 50-800 mg/day, therefore the comparison is again difficult to interpret;

Lines 205-206- Please specify the units for QTc, which is „ms”, I presume;

Line 293- Which were the strict inclusion criteria? Because in lines 89-90, it looks like only the age, the ICU admission, and the prescription of an antipsychotic are mentioned in this category, the other criteria are exclusionary.

Author Response

Reviewer 1

Comment 1: Thank to the Authors for considering my recommendations, but there are a few aspects that require further clarification.

Response 1: We really appreciate the reviewer for their time in reviewing our manuscript. All their comments have clearly improved our manuscript.

Comment 2: Lines 23 and 184-185- Please specify the units used for the measurement of the initiation dose;

Response 2: Thank you for the comment. The dose unit is “milligrams” and was specified in Lines 22 and 188-189 as requested.

Comment 3: Where are the figures 2a and 2b? In the manuscript, there is only one figure 2 and two figures 3 (a and b).

Response 3: we apologize for this mistake. We have corrected it in the manuscript.

Comment 4: Please remove the symbol ¶ after  A and B in Figure 3.

Response 4: We have removed it as requested.

Comment 5: Line 96- What does „received different strengths of the same medication” mean? Did the Authors refer to variations in the daily doses in the same patient during the ICU stay? Or maybe to the patients who received different daily doses prior to the inclusion in the current study? Please clarify.

Response 5: Thank you for this question. We refer to those who were tried on multiple strengths (tablets) due to absence of improvement in their symptoms (for example started on 50mg tablets then discontinued and started on 100 mg tablets to reduce pill burden…etc) despite doing that within 72 hours of the 1st medication initiation or beyond that time period. Changing a strength within 72 hrs doesn’t give a medication enough time to exert its full efficacy on the long terms. On the other side, changing the strength after 72 hrs might led to speculating a potential impact of the previous dose beside the problem in calculating the true duration of medication exposure considering the used strength and the mean total daily dose throughout the whole administration time. Therefore, those who received multiple strength prior to improvement in their symptoms were excluded while those who remained on the same strengths were included. We have added “prior to delirium improvement” in Line 98 to clarify that. Also, it was explained in details in Figure 1.

Comment 6: Table 1- What type of psychiatric disorders did the patients present?

Response 6: Thank you for this question. The patient in haloperidol group was having a history of anxiety history, while the patient treated with quetiapine was having a history of mixed anxiety and depression.

Comment 7: Table 2- The comparison between the four antipsychotics is difficult to interpret, because only 4.3% and 8.5% of the patients received olanzapine and risperidone, respectively; this is a limitation of the study that should be mentioned in the dedicated paragraph;

Response 7: we appreciate the reviewer for this comment. We totally agree with that.  We have included that in the paragraph discussing the limitations of our study and made some adjustment based on that.

Comment 8: Lines 174-176- Why were some patients initiated on antipsychotics after ICU admission but prior to the delirium diagnosis? Did they have psychotic disorders or other pathology requiring antipsychotics? According to the exclusion criteria in lines 90-97, patients who were not delirious were not included in the analysis. Clarifying this aspect is important since 21.4% of the haloperidol-treated patients and 71.4% of the quetiapine-treated patients were in this category.

Response 8: We appreciate the reviewer for these questions. As demonstrated in Figure 1, the method that was used to recruit patients in our study was via ICU admission and administration of medications only. Having said that, patients who were “initiated on antipsychotics after ICU admission but prior to the delirium diagnosis” were clinically not delirious and no clear diagnosis nor assessment was documented for them. Despite that, they did develop delirium which was dated and documented based on CAM-ICU assessment. In fact, this raises questions regarding the prophylactic efficacy of such medications, if the intent of use was that (given the risk factors they had). Although 21.4% of the haloperidol-treated patients and 71.4% of the quetiapine-treated patients were in this category, that difference was not significant, hence its impact as a covariate on delirium was not evaluated.

Comment 9: Lines 185-193- those differences between the initial and maintenance doses of haloperidol and quetiapine, respectively,  may be related to the distinct pharmacology of the two antipsychotics, because haloperidol is a strong antagonist of D2 receptors and quetiapine only a weak D2-antagonist; increasing the dose of quetiapine is perfectly explainable because at low doses it is only used as an off-label hypnotic (25-200 mg/day) -https://pubmed.ncbi.nlm.nih.gov/22510671/; also, the admitted interval of doses for haloperidol according to SPC is 5-20 mg/day, and for quetiapine 50-800 mg/day, therefore the comparison is again difficult to interpret;

Response 9: We really appreciate the reviewer for this clarification. We have highlighted and discussed that in lines 283-294

Comment 10: Lines 205-206- Please specify the units for QTc, which is „ms”, I presume;

Response 10: We appreciate the reviewer for this comment. This is true and we have incorporated that as requested.  

Comment 11: Line 293- Which were the strict inclusion criteria? Because in lines 89-90, it looks like only the age, the ICU admission, and the prescription of an antipsychotic are mentioned in this category, the other criteria are exclusionary.

Response 11: We appreciate the reviewer for this comment. We have modified that to the “the exclusion criteria….”. Please refer to line 311.

Reviewer 2 Report

Comments and Suggestions for Authors

I thank the Authors for their revision, which discreetly improved the quality of their manuscript. However, please;

- remove the abbreviations from Figure 1 (or at least explain them);

- briefly discuss the implications of Covid-19 on their delirium in the three patients mentioned and assess whether they differed from the others in specific features; 

Author Response

Reviewer 2

Comment 1: I thank the Authors for their revision, which discreetly improved the quality of their manuscript.

Response 1: We really appreciate the reviewer for all the comments that have clearly improved our manuscript. We sincerely appreciate their valuable time in doing that.  

Comment 2: Remove the abbreviations from Figure 1 (or at least explain them)

Response 2: We appreciate the reviewer for this comment. We have removed all abbreviations (such as ICU, CAM-ICU, hrs, n) as well as symbols (such as #) and updated figure 1 accordingly as requested.

Comment 3: Briefly discuss the implications of Covid-19 on their delirium in the three patients mentioned and assess whether they differed from the others in specific features.

Response 3: We appreciate the reviewer for this comment. We would like from the reviewer to excuse us from responding to this comment as another project on delirium among COVID-19 patients is currently conducted by our team.

Reviewer 3 Report

Comments and Suggestions for Authors

Study design and sample size are not appropriate to make conclusion

Author Response

Reviewer 3

Comment 1: Study design and sample size are not appropriate to make conclusion

Response 1: We appreciate the reviewer for this comment. We have stated that as limitations of the study and because of that we have stated in the conclusion that “However as with other antipsychotics, its use would be warranted based on its psycho-pharmacological properties”.